# Hyperspectral reflectance spectra of floating matters derived from HICO observations

Chuanmin Hu

College of Marine Science, University of South Florida, St. Petersburg, Florida, 33701, USA

*Correspondence to:* Chuanmin Hu (huc@usf.edu)

**Abstract**

Using data collected by the Hyperspectral Imager for the Coastal Ocean (HICO) on the International Space Station between 2010 – 2014, hyperspectral reflectance of various floating matters in global oceans and lakes are derived for the spectral range of 400 – 800 nm. Specifically, the entire HICO archive of 9,411 scenes is first visually inspected to identify suspicious image slicks. Then, a nearest-neighboring atmospheric correction is used to derive surface reflectance of slick pixels. Finally, a spectral unmixing scheme is used to derive the reflectance spectra of floating matters. Analysis of the spectral shapes of these various floating matters (macroalgae, microalgae, organic particles, whitecaps) through the use of a Spectral Angle Mapper (SAM) index indicates that they can mostly be distinguished from each other without the need of ancillary information. Such reflectance spectra from the consistent 90-m resolution HICO observations are expected to provide spectral endmembers to differentiate and quantify the various floating matters from existing multi-band satellite sensors and future hyperspectral satellite missions such as NASA's Plankton, Aerosol, Cloud, and ocean Ecosystem (PACE) mission and Surface Biology and Geology (SBG) mission.

**Keywords:** Remote sensing, hyperspectral, HICO, OCI, PACE, SBG, floating matters, *Ulva, Sargassum, Noctiluca, Trichodesmium, Microcystis,* brine shrimp, oil slicks, whitecaps, marine debris.

## 1. Introduction

Since the debut of the first proof-of-concept Coastal Zone Color Scanner (CZCS, 1978 – 1986), satellite ocean color missions have evolved from the original goal of mapping phytoplankton biomass and primary production to many other applications. Because of improved spectral resolution and instrument sensitivity, mapping various floating matters also becomes possible (IOCCG, 2014). These floating matters range from living to non-living, including *Sargassum* macroalgae, *Ulva* macroalgae, cyanobacterium *Microcystis*, cyanobacterium *Trichodesmium*, dinoflagellate *Noctiluca*, aquatic plants, brine shrimp cysts, oil slicks, pumice rafts, sea snots, marine debris, among others (Qi et al., 2020; Hu et al., 2022).

Currently, mapping floating matters using optical remote sensing requires the detection of a spatial anomaly using the near-infrared (NIR) bands, and then discrimination of the anomaly by comparing its spectral characteristics with known spectra of floating matters (Qi et al., 2020), or by using ancillary information (e.g., in certain regions a spatial anomaly can only be caused by a certain type of floating algae). Spectral discrimination requires the knowledge of

spectral signatures of various floating matters. However, despite scattered laboratory or field measurements of certain
types of floating matters, hyperspectral data of these floating matters are mostly unavailable. Although medium-
resolution (300-m) sensors such as the Ocean and Land Colour Imager (OLCI) has been used to show spectral
variations of floating matters (Qi et al., 2020), the data are not hyperspectral, therefore certain spectral features may
have been missed. For example, various pigments (e.g., chlorophyll-*a*, *b*, *c*, Fucoxanthin, Zeaxanthin, phycocyanin,
carotenoid, etc.) have been found in natural populations of microalgae (i.e., phytoplankton, Bidigare et al., 1990;
Bricaud et al., 2004) and macroalgae (e.g., Bell et al., 2015; Wang et al., 2018). These pigments often have narrow
absorption and reflectance features that can be missed by multi-band sensors, therefore requiring more spectral bands
or hyperspectral data to perform spectroscopic analysis.
Data collected by the Hyperspectral Imager for the Coastal Ocean (HICO) on the International Space Station may
serve for this purpose. HICO has 128 bands covering a spectral range of 353 – 1080 nm. From its entire mission of
2010 – 2014, a total of > 10,000 scenes have been collected at a spatial resolution of about 90 m, each containing
about $512 \times 2000$ pixels. On average, only 6 scenes were collected per day around the globe, mostly over land and
coastal waters. Because of its stable calibration (Ibrahim et al., 2018) and relatively high signal-to-noise ratios (Hu et
al., 2012), deriving hyperspectral surface reflectance of water targets should be feasible. Indeed, after vicarious
calibration and atmospheric correction, hyperspectral reflectance data over water have been generated (Ibrahim et al.,
2018) and made available through the NASA OB.DAAC (https://oceancolor.gsfc.nasa.gov). However, these data
products are not applicable to image pixels containing floating matters due to their interference with the atmospheric
correction scheme.
The primary objective of this paper is to derive HICO-based hyperspectral reflectance of various floating matters.
This requires customized atmospheric correction and pixel unmixing to account for the small proportion of floating
matters within an image pixel. From such derived spectra, a secondary objective is to analyze whether they can be
differentiated spectrally. Similar to the compiled hyperspectral dataset for inherent and apparent optical properties to
support future hyperspectral missions such as NASA's Plankton, Aerosol, Cloud, and ocean Ecosystem (PACE)
mission (Casey et al., 2020), such a dataset for floating matters is expected to help develop or improve algorithms for
the PACE mission as well as for the hyperspectral Surface Biology and Geology mission currently being planned by
NASA (Cawse-Nicholson et al., 2021).
**2. Data and Methods**
HICO Level-1B (calibrated radiance) data were obtained from the NASA Goddard Space Flight Center
(https://oceancolor.gsfc.nasa.gov). Of the total collected >10,000 scenes, 9,411 were available through this data portal.
They were all downloaded, and the following 4 steps were used to derive spectral reflectance of various floating
matters.
Step 1 is to generate quick look Red-Green-Blue (RGB) and False-color RGB (FRGB) images with Rayleigh corrected
reflectance ($R_{rc}$, dimensionless) in three HICO bands using the same methods as in Qi et al. (2020) and in the NOAA
OCview online tool (Mikelsons and Wang, 2018). In the FRGB images, a near-infrared (NIR) band is used to represent
the green channel, thus making floating matters often appear greenish due to their elevated NIR reflectance. Here, $R_{rc}$
was generated using the NASA software SeaDAS (version 7.5). Mathematically, it is derived as
$$R_{rc} = (R_t - R_r)/(t\, t_o\, t_{O2}\, t_{H2O}),$$
$$R_t = \pi\, L_t^{*} / F_o \cos(\theta_o),$$
$$R_r = \pi\, L_r / F_o \cos(\theta_o), \tag{1}$$
where $L_t^{*}$ is the at-sensor total radiance after vicarious calibration and adjustment of two-way gaseous absorption (e.g.,
Ozone), $L_r$ is at-sensor radiance due to Rayleigh scattering, $F_o$ is the extraterrestrial solar irradiance, $\theta_o$ is the solar
zenith angle, $t$ is the diffuse transmittance from the image pixel to the satellite, $t_o$ is the diffuse transmittance from the
sun to the image pixel, $t_{O2}$ and $t_{H2O}$ are the two-way transmittance due to absorption by atmospheric $O_2$ and $H_2O$,
respectively. For simplicity, the wavelength dependency is omitted here.
Step 2 is to determine image slicks through visual inspection of both RGB and FRGB images. Fig. 1a shows an FRGB
image captured in the central western Atlantic, where an elongated greenish slick is identified.
Step 3 is to derive surface reflectance ($R$, dimensionless) of the slick pixels (i.e., those containing floating matters)
*and* nearby water pixels. While the latter is straightforward because $R$ at each pixel is a standard output of the SeaDAS
software, the former is problematic because standard atmospheric correction in SeaDAS fails over floating matters
due to their elevated NIR reflectance. Such elevated NIR reflectance violates the atmospheric correction assumptions
(i.e., negligible reflectance in the NIR, or fixed relationships between the red and NIR wavelengths) for slick pixels.
Therefore, a nearest-neighbor atmospheric correction (Hu et al., 2000) was used to estimate $R$ of the slick pixels.
Specifically, from the SeaDAS output of $R_{rs}$, we have
$$R = \pi\, R_{rs} = (R_t - R_r - R_a)/(t\, t_o\, t_{O2}\, t_{H2O}), \tag{2}$$
where $R_{rs}$ is the surface remote sensing reflectance ($sr^{-1}$), $R_a$ is the at-sensor aerosol reflectance (and reflectance due
to aerosol-molecule interactions as well as due to sun glint and whitecaps). The difference between $R$ and $R_{rc}$ in Eqs.
(2) and (1), respectively, is the removal of $R_a$ in (2). Estimation of $R_a$ at each pixel represents the "core" of any
atmospheric correction scheme. The SeaDAS estimation of $R_a$ is valid over water pixels, but not valid over the slick
pixels. Therefore, $R_a$ over water pixels was used as a surrogate to represent $R_a$ over the nearby slick pixels, from which
$R$ over slick pixels was derived. This is why such an approach is called "nearest-neighbor" atmospheric correction
(Hu et al., 2000). In this context, the slick pixel is called "target", and the nearby water pixel is called "reference".
Their surface reflectance are called $R^T$ and $R^R$, respectively. Fig. 1b shows examples of $R^T$ and $R^R$.






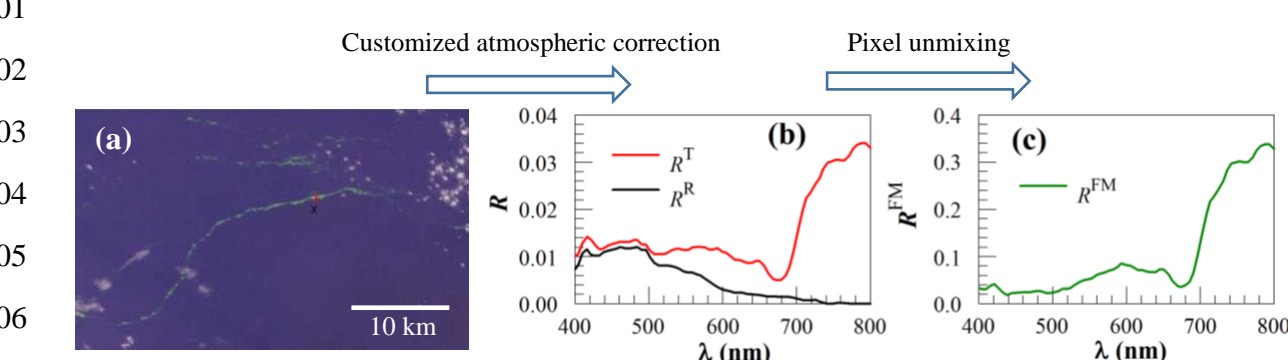

107

**Figure 1. Demonstration of how surface reflectance of floating matter ($R^{FM}$) is derived. (a) FRGB image on 1 July 2012 showing several greenish image slicks in the Amazon River plume. The image covers a region of about 40 km × 24 km, with the "Target" (6.65914°N, 51.2395°W) and "Reference" (6.64847°N, 51.2411°W) pixels marked with a red "×" and a black "×", respectively. (b) Their corresponding $R^T$ and $R^R$, with the latter derived from SeaDAS and the former derived from a nearest-neighbor atmospheric correction. (c) $R^{FM}$ derived from $R^T$ and $R^R$ using Eq. (4), with χ being estimated to be 10%.**

The final step, Step 4, is to perform spectral unmixing of $R^T$. This is because floating matters often cover only a small portion a pixel (Hu, 2021a). In this step, the derived $R^T$ from Step 3 is assumed to be a linear mixture of two endmembers: floating matter ($R^{FM}$) and water ($R^W$):

$$R^T = \chi R^{FM} + (1 - \chi)R^W = \chi R^{FM} + (1 - \chi)R^R \qquad (3)$$

Here, χ is the subpixel portion of floating matter which can vary between 0.0% and 100%, $R^W$ is assumed to be $R^R$. Then, the final product, $R^{FM}$, is derived as

$$R^{FM} = R^R + (R^T - R^R)/\chi \qquad (4)$$

In the right-hand side of Eq. (4), the only unknown is χ. In practice, assuming $R^{FM}$ at 750 nm ≈ 0.3 as revealed by independent measurements of floating macroalgae (Hu, L. et al., 2017; Wang et al., 2018), χ is estimated through linear unmixing as

$$\chi = [R^T(754) - R^R(754)]/[0.3 - R^R(754)] \qquad (5)$$

Here, with $R^T(754)$ varying between $R^R(754)$ and 0.3, χ ranges between 0.0% and 100%. Plugging this mixing ratio into Eq. (4) will derive $R^{FM}$. Fig. 1c shows the example of how $R^{FM}$ is derived from $R^T$ and $R^R$ of Fig. 1b once they are known from Step 3, with χ being estimated to be 10%.

Once $R^{FM}$ is derived, a spectral angle mapper index (SAM, Kruse et al., 1993) was used to determine whether different floating matters were spectrally different. SAM was used because it is based on spectral shape only. SAM is the angle between two spectral vectors, defined as (Kruse et al., 1993):

$$SAM \text{ (degrees)} = \cos^{-1}[(\sum x_i y_i) / (\sqrt{\sum x_i^2} \sqrt{\sum y_i^2})]. \qquad (6)$$

Here, $x$ and $y$ represent two spectral vectors with the $i^{th}$ band from 1 to $N$. An SAM of $0^o$ indicates identical spectral
shapes between $x$ and $y$ regardless of their difference in magnitudes, while an SAM of $90^o$ indicates completely
different spectral shapes. An SAM of $< 5^o$ indicates that the two spectra are very similar (Garaba and Dierssen, 2018).
**3. Results: HICO reflectance spectra of floating matters**
The approach above was applied to the visually identified image slicks to derive $R^{FM}(\lambda)$. These include: 1) *Sargassum*
*fluitans/natans* in the Atlantic (including the Caribbean Sea and Gulf of Mexico), 2) *Ulva prolifera* in the western
Yellow Sea (near Qingdao, China), 3) *Kelp* in South Atlantic, 4) *Trichodesmium* around Australia, in the Gulf of
Mexico and Persian Gulf, in the South Atlantic Bight, Bay of Bengal, near Hawaii and Pagan Island (middle Pacific),
5) Cyanobacteria of *Microcystis* in Taihu Lake, Lake Woods, and Lake of Victoria, 6) Red *Noctiluca scintillas* (*RNS*)
in the East China Sea, and coastal waters off Japan, 7) Brine shrimp cysts in the Great Salt Lake, 8) Oil slicks in the
Gulf of Mexico, 9) Whitecaps (foam) in the Arabian Sea, Caspian Sea, and Bohai Sea, 10) Ice in Lake Baykal, 11)
some unknown algae features. For convenience, they are grouped into 4 figures: Fig. 2 for macroalgae (*Sargassum*,
*Ulva*, and kelp), Fig. 3 for microalgae (*Trichodesmium*, *Microcystis*, red *Noctiluca scintillas* or *RNS*), Fig. 4 for organic
particles and ocean/lake bubbles, and Fig. 5 for unknown algae scums.

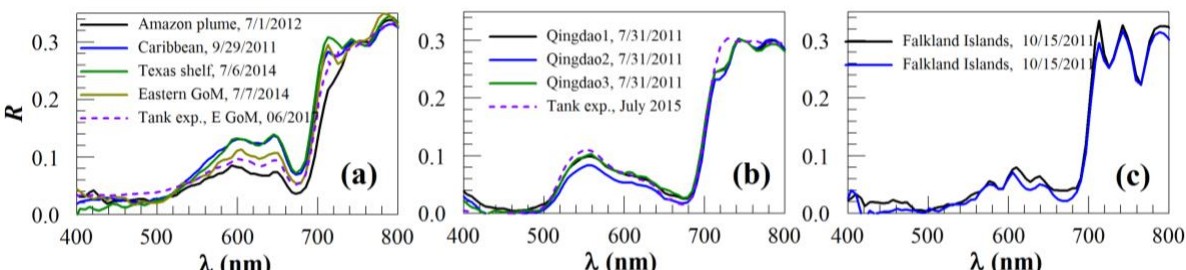

**Figure 2: Surface reflectance ($R$, dimensionless) of macroalgae: (a) pelagic *Sargassum fluitains/natans*, (b) *Ulva prolifera*, (c)**
**kelp. The dashed lines in (a) and (b) denote $R$ from water tank experiments of Wang et al. (2018) and Hu, L. et al. (2017),**
**respectively.**


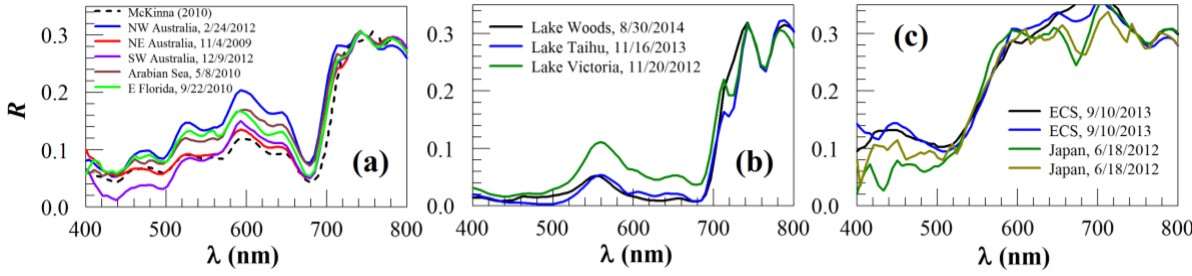


**Figure 3. Surface reflectance ($R$, dimensionless) of floating scums of microalgae: (a) *Trichodesmium*, (b) *Microcystis*, (c)**
**red *Noctiluca* near Yangtze River of the East China Sea and in Sagami Bay of Japan. The dashed line in (a) denote field**
**measured $R$ by McKinna (2010).**


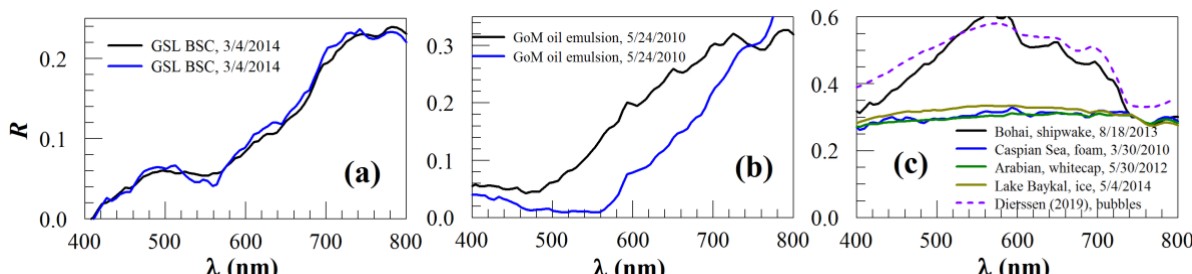


**Figure 4: Surface reflectance (*R*, dimensionless) of various floating materials: (a) Brine shrimp cysts in the Great Salt**
**Lake (GSL), (b) emulsified oil from the Deepwater Horizon oil spill, and (c) shipwake, seafoam, whitecap and ice. The**
**dashed line in (c) denotes submersed bubbles measured by Dierssen (2019), which is similar to the shipwake spectrum.**
**Note the similarity among other spectra.**




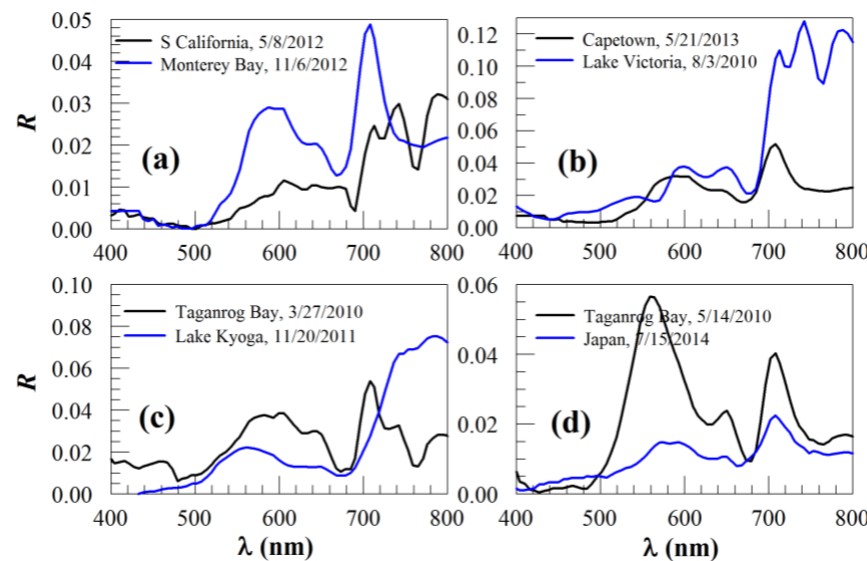







**Figure 5: Surface reflectance (*R*, dimensionless) of known and unknown algae scums. (a) Blooms off southern California**
**and in Monterey Bay that are thought to be *Lingulodinium polyedrum* (Cetinic, 2009) and *Akashiwo sanguinea* (Jessup et**
**al., 2009), respectively. (b) Blooms of unknown types of algae off Cape Town (South Africa) and in Lake Victoria, both**
**likely to be dinoflagellates. Note the different spectra shape of the Lake Victoria bloom as compared with the cyanobacterial**
**bloom in the same lake (Fig. 3b). (c) Blooms of unknown types of algae in Taganrog Bay and Lake Kyoga. (d). Blooms of**
**unknown types of algae in Taganrog Bay (note the difference from Fig. 5c) and in Japan coastal waters.**
Of all spectra presented in Figs. 2 – 4, one common feature for all floating macroalgae and microalgae (except red
*Noctiluca*) is the red-edge reflectance (i.e., the sharp increase from about 670 nm to the NIR wavelengths). Such a
common feature is due to both chlorophyll-*a* absorption around 670 nm and high reflectance in the red and NIR
wavelengths due to macroalgae mats or microalgae scums (Kazemipour et al., 2011; Launeau et al., 2018). The lack
of such a red-edge feature in some of the red *Noctiluca* reflectance spectra (Fig. 3c) is possibly due to the lack of
chlorophyll-*a* pigment because red *Noctiluca* is heterotrophic (i.e., it does not contain pigments unless it feeds on other
algae). Other than the common red-edge reflectance, the contrasting spectral shapes of the various types of floating
macroalgae and microalgae are due to their different pigment compositions (see below). In contrast, the non-leaving
floating matters do not show red-edge reflectance or other pigment-induced spectral features in the visible wavelengths
(Fig. 4). In Fig. 5, in addition to pigment absorption, high scattering due to high concentrations of algae particles
together with sharp increases of water absorption from the red to the NIR wavelengths lead to the local reflectance
peak around 700 nm (Fig. 5) and, depending on the particle concentrations, the peak wavelength may be slightly
shifted, for example from 700 to 710 nm.
**4. Discussion**
**4.1. Uncertainties in the derived RFM**
There are several assumptions used in the nearest-neighbor atmospheric correction and spectral unmixing (Eq. 4).
Violations of these assumptions will cause errors in the derived $R^{FM}$ spectra. For example, if the atmosphere over the
floating matter pixel is different from over the nearby water, the nearest-neighbor atmospheric correction may not be
applicable. In practice, however, because the target and reference pixels are very close (< 1 km), such a violation is
unlikely. In Step 4, the water within the FM-containing pixel is assumed to be the same as the nearby water. Because
of the close proximity of the two pixels, this assumption should be valid for most cases unless the FM-containing pixel
is at an ocean front where different water masses converge. The departure of $R^{FM}(754)$ from the assumed 0.3 will also
lead to errors in the estimated $\chi$ (and therefore $R^{FM}$). However, as long as $R^{W}$ (i.e., $R^{R}$) in Eqs. (4) & (5) is $<< R^{FM}$, the
shape of $R^{FM}$ is still retained, although the magnitude departs from the "truth" in proportional to the departure of
$R^{FM}(754)$ from 0.3. Indeed, the condition of $R^{W} << R^{FM}$ can be satisfied for $\lambda > 600$ nm for most floating matters unless
the water is extremely turbid. Even for turbid waters, for certain floating matters where $R^{FM}$ is elevated at $\lambda > 530$ nm
(e.g., red *Noctiluca*, brine shrimp cysts, ice), the shape of the derived $R^{FM}$ should still be valid for $\lambda > 530$ nm. Indeed,
when $R^{W}$ is $<< R^{FM}$, even a simple subtraction of $R_{rc}$ or TOA radiance between the target pixel and reference pixel, as
demonstrated in Gower et al. (2006), may retain the spectral shapes of floating matters.
Another uncertainty source can come from the assumption of linear mixing between floating matters and water (Eq.
(3)). For macroalgae, the linear mixing up to the reflectance saturation level has been shown in laboratory experiments
(Hu. L et al., 2017; Wang et al., 2018). As long as the macroalgae stay on the very surface of water (as opposed to be
submerged under the surface), this assumption should be valid not just for macroalgae but for all floating matters. For
the same reason, if certain portions of kelp are submerged in water, large uncertainties may result from the linear
unmixing scheme. Under high-wind conditions, the strong mixing may result in submerged algae (especially for
microalgae), thus violating the linear mixing rule. However, the cases presented in Figs. 2 - 5 were selected very
carefully to avoid high wind speed (> 5 m s$^{-1}$, where wind speed was obtained from the National Centers for
Environmental Prediction). Therefore, such mixing induced uncertainties are unlikely.
Additional uncertainties may come from the HICO radiometric calibration, which affects $R_t$ and all derivative products.
Through the use of the Marine Optical Buoy (MOBY) and other clear-water sites, HICO has been calibrated
vicariously (Ibrahim et al., 2018), which resulted in significant improvements in the retrieved $R_{rs}$ over water as
compared with data without vicarious calibration. However, after the vicarious calibration, while the spectral shape
of $R_{rc}$ over water appears correct, the shape of $\Delta R_{rc}$ over land appears to be biased low at $\lambda > 800$ nm. Without vicarious
calibration, the opposite is observed. This is possibly due to the non-linear effects in the detector response to incoming
light, and currently there appears no reliable way to address this issue (A. Ibrahim, personal comm.). Similarly,
calibration for $\lambda < 450$ nm may be subject to larger errors than for $\lambda$ between 450 and 800 nm. Therefore, $R^{FM}$ in the
range of 800 – 900 nm is omitted here, and interpretation of 400 – 450 also requires more caution. Similarly, the
spectral wiggling between 700 and 800 nm (e.g., Fig. 3b) appears to come from residual errors in correcting water
vapor absorption and oxygen absorption in the atmosphere. Therefore, although the spectral wiggling does not affect
the overall shape of the red-edge reflectance, it may not be used for algorithm development to discriminate floating
matter types.
Indeed, with all these possible sources of uncertainties, such HICO-derived $R^{FM}$ can still be used for spectral
discrimination of different floating matters without ambiguity, as shown below.

**4.2. Implications for spectral discrimination**

Spectral discrimination can be performed through either visual inspection or the use of certain type of similarity index
(e.g., SAM, Eq. 6). Here, results of the SAM analysis are presented in Table 1, followed by descriptions of visual
inspection to interpret the spectral similarity or difference. Because nearly all floating algae show typical red edge
reflectance, discrimination of different algae type is focused on wavelengths < 670 nm. To discriminate floating algae
from non-living floating matters (e.g., marine debris), on the other hand, the inclusion of 670 nm is critical.
Furthermore, because HICO data are noisy for wavelengths < 450 nm, the SAM calculation was restricted to 450 –
670 nm from most $R^{FM}$ spectra of Figs. 2 – 4.
Table 1 shows the SAM results for three types of macroalgae (*Sargassum*, *Ulva*, kelp), three types of microalgae
(*Trichodesmium*, *Microcystic*, red *Nocticula scintillas* or *RNS*), and one type of organic matter (brine shrimp cysts or
*BSC*). **Here, unless noted, *Sargassum* refers to *Sargassum fluitans/natans* (dominant pelagic type in the Atlantic**
**ocean) and *Ulva* refers to *Ulva prolifera* (dominant pelagic type in the Yellow Sea).** For the same floating matter,
if field-based $R^{FM}$ is available, then it is used as the reference, otherwise the mean HICO-derived $R^{FM}$ is used as the
reference. For SAM between different floating matters, all HCIO-derived $R^{FM}$ from both types are used (e.g., 4
*Sargassum* $R^{FM}$ of Fig. 2a and 3 *Ulva* $R^{FM}$ of Fig. 2b are used to calculate 12 SAM values), with their mean and
standard deviations listed in Table 1.

**Table 1. Spectral Angle Mapper values (degrees) between different floating matters for the spectral range of 450 – 670 nm, derived from the HICO-derived and field-measured spectra shown in Figs. 2-4. An SAM of 0° indicates identical spectral shape, while an SAM of 90° indicates completely different spectral shape.** *Sarg*: *Sargassum fluitans/natans*; *Ulva: Ulva prolifera*; *Tricho*: *Trichodesmium*; *Micro*: *Microcystis*; *RNS*: red *Noctiluca scintillas*; *BSC*: brine shrimp cysts. **Because all floating algae show similar red-edge reflectance with a reflectance trough around 670 nm, the exclusion of wavelengths of > 670 nm is to reduce the similarity among different types of floating algae.**

| | Sarg | Ulva | Kelp | Tricho | Micro | RNS | BSC |
|---|---|---|---|---|---|---|---|
| *Sarg* | **4.5±1.6** | | | | | | |
| *Ulva* | 27.2±2.5 | **2.9±0.5** | | | | | |
| *Kelp* | 13.7±1.8 | 32.5±1.3 | **2.7±0.4** | | | | |
| *Tricho* | 15.4±4.6 | 25.1±2.0 | 23.1±3.2 | **2.8±2.0** | | | |
| *Micro* | 32.9±7.5 | 16.8±5.6 | 39.0±7.7 | 28.8±5.1 | **4.6±2.5** | | |
| *RNS* | 9.9±2.4 | 31.4±2.8 | 16.7±3.0 | 17.2±2.1 | 34.7±6.7 | **1.8±0.7** | |
| *BSC* | 20.7±0.9 | 39.3±2.4 | 27.0±3.1 | 21.2±1.6 | 40.9±5.5 | 14.5±3.1 | **1.1±0.0** |
| | *Sarg* | *Ulva* | *Kelp* | *Tricho* | *Micro* | *RNS* | *BSC* |

257

For each type of floating matter, HICO-derived $R^{FM}$ is very similar to either field-measured $R^{FM}$ or to their mean $R^{FM}$, with SAM < 4.6°. In contrast, SAM between different floating matters is always > 9.9°. These results suggest that, if these floating matters represent all that can be found in natural waters, they can be differentiated through spectroscopy analysis without any other ancillary information (e.g., knowledge of local oceanography or dominant floating algae type). This is despite the possible uncertainties in their reflectance magnitude, as discussed above. In the natural environments, however, there may be other types of floating algae whose spectral shapes may be similar to *Sargassum fluitans/natans* (e.g., *Sargassum honeria* in the East China Sea *or* other brown algae) or *Ulva prolifera* (e.g., other green algae). Therefore, some form of ancillary information in addition to spectroscopy is still required in order to differentiate floating algae type.

The results from the SAM table can also be explained through visual inspection and interpretation of the spectral shapes, as discussed below.

From Fig. 2, it is clear that although the three types of macroalgae all share the same red-edge reflectance in the NIR, they have different spectral shapes in the visible wavelengths. Unlike the *Ulva* reflectance with a local peak around 560 nm, the spectral shapes of *Sargassum* reflectance resemble those of typical brown algae where the local reflectance trough around 625 nm is induced by chlorophyll-*c* absorption and the low reflectance below ~520 nm is due to carotenoid pigment absorption. These characteristics make it easy to distinguish *Sargassum* from *Ulva* (SAM > 27°, Table 1). On the other hand, it appears more difficult to spectrally discriminate *Sargassum* from kelp because they

both have reference peaks around 600 – 645 nm, and because they also share a common reflectance trough around
625 nm. However, considering *Sargassum* is moving in the ocean while kelp is fixed in location, they can be separated
using sequential images. Even from a single image, when most visible wavelengths are used, *Sargassum* and kelp can
still be spectrally discriminated (SAM > 13°, Table 1). Within the group of *Sargassum* spectra (Fig. 2a), there is some
variability in the magnitude between 560 – 700 nm. It is unclear what caused such variability, although it could be
due to changes in carbon to chlorophyll ratio in *Sargassum* of different environment, as observed from kelp (Bell et
al., 2015). Such a variability, however, would not impact the spectral discrimination of *Sargassum* against other
floating matters, as SAM between *Sargassum* spectra is < 5°, much lower than between *Sargassum* and any other
floating matters (Table 1).
Similar to the macroalgae, the microalgae scums also show elevated NIR reflectance (Fig. 3), and their spectral shapes
in the visible make them straightforward to distinguish from each other (SAM > 17°), and also straightforward to
distinguish from macroalgae (SAM > 9.9°). One exception may be the cyanobacterial scums (blue-green algae blooms)
(Fig. 3b) as they show reflectance peak around 550 nm, similar to *Ulva* (Fig. 2b). However, reflectance around 550
nm is nearly symmetric for cyanobacterial scums, but asymmetric for *Ulva*. There is also a local reflectance trough
around 625 nm for cyanobacterial scums due to absorption of phycocyanin, but such a trough is lacking in the *Ulva*
spectra. Such characteristic makes it possible to differentiate between the two even without *a priori* knowledge of the
ocean or lake environment, as the SAM between the two groups is ~16.8° (Table 1). What's interesting is that within
each class, either *Trichodesmium* or *Microcystis*, although the spectral shape is nearly identical from different spectra
(SAM < 5°), there is substantial variability in the magnitude in the visible wavelengths, which might be due to changes
in their carbon to chlorophyll ratios (Behrenfeld et al., 2005). Furthermore, the spectral wiggling features between 450
and 660 nm in Fig. 3a are due to *Trichodesmium*-specific pigments such as phycourobilin, phycoerythrobilin, and
phycocyanin that absorb light strongly at 495, 550, and 625 nm, respectively (Navarro Rodriguez, 1999). These
features are unique to *Trichodesmium* scums, which make it straightforward to develop classification algorithms once
certain spectral bands are available to capture these features (e.g., Hu et al., 2010).
Of all microalgae scums of Fig. 3, the spectral shapes of red *Noctiluca* (Fig. 3c) appear different from all others, but
they show the same characteristics as reported from the limited field measurements (Van Mol et al., 2007): a sharp,
featureless increase from ~520 nm to ~600 nm. This unique spectral shape makes *RNS* different from all other floating
matters (SAM > 9.9°, Table 1). The difference within this group is that the spectra from Sagami Bay off Japan show
reflectance troughs around 670 nm. Because red *Noctiluca* is known to feed on other algae, it is speculated that the
670-nm trough is due to chlorophyll pigments of the consumed algae. Once more hyperspectral data are available in
the future to test this hypothesis using field data, this characteristic may be used to study how red *Noctiluca* interacts
with other algae. On the other hand, once more hyperspectral data are available in the future, it is also possible to test
whether other algae (e.g., *Mesodinium rubrum*, Dierssen et al., 2015), once forming surface scums, have similar
spectral shapes as those of red *Noctiluca*.
The non-algae floating matters in Fig. 4 show spectral characteristics different from both macroalgae and microalgae,
for example they lack the typical red-edge reflectance of vegetation, and lack of typical spectral variations in the

visible wavelengths due to pigment absorption. Within this group, the organic matters of BSC (Fig. 4a) and emulsified oil (Fig. 4b) show some degrees of similarity as they also have monotonic reflectance increases from a wavelength between 500 – 560 nm to at least 740 nm. The difference between them is that BSC reflectance starts to increase always at ~560 nm with an inflection wavelength ~640 nm, while reflectance of oil emulsions start to increase at variable wavelengths without any inflection between 560 – 740 nm. Indeed, the infection at ~640 nm appears to be a common feature between BSC slicks and coral spawn slicks (Yamano et al., 2020). In contrast, depending on the oil emulsion state, oil emulsion may have different spectral characteristics (Lu et al., 2019), suggesting that there is no fixed "endmember" spectra for oil spills.

The inorganic "particles" (i.e., water bubbles, ice) also have distinctive spectral shapes. The examples in Fig. 4c indicate that submersed bubbles from shipwakes are similar in spectral shapes, but all others are nearly identical in their lack of any spectral features. Rather, foams, whitecaps, and ice all show flat reflectance spectral shapes between 400 – 800 nm that are consistent with *in situ* measurements of foams (Dierssen, 2019). The lack of spectral features is similar to marine debris (Garaba and Dierssen, 2020). Such a similarity will make detection of marine debris very difficult, especially around ocean fronts because these are where surface materials tend to aggregate and foams also tend to form.

In addition to the spectra of Figs. 2-4 that can be well recognized, HICO also showed reflectance spectra that are difficult to discriminate from spectroscopy alone, as shown in Fig. 5. Without a known reflectance library, one can only speculate what algae type could be responsible for the algae scum spectra from some ancillary information in the literature. For example, the often-reported blooms of *Lingulodinium polyedrum* and *Akashiwo sanguinea* in coastal waters off southern California and in Monterey Bay, respectively, may show spectral shapes of Fig. 5a when they are heavily concentrated in surface waters. Inference may also be made for other cases once similar ancillary information is available. Even when such information is absent, one can still rule out some possibilities simply based on the spectral shapes. For example, the reflectance spectrum in Fig. 5b from Lake Victoria cannot be from cyanobacteria that has been often reported in this lake (Fig. 3b), but it is most likely from a dinoflagellate bloom, as blooms of other algae types have also been reported in this lake (Haande et al., 2011). Likewise, the different spectra from the same Taganrog Bay in Figs. 5c & 5d suggest different algae type. Clearly, although cyanobacterial blooms have been reported in many lakes, without spectral diagnosis one cannot simply jump to the conclusion that a freshwater bloom is caused by a certain type of cyanobacterium.

### 4.3. Implications for current and future satellite missions

Because HICO is a pathfinder sensor that collected only a limited number of scenes, not all reported floating matters have been captured. For example, no HICO scene appears to have captured pumice rafts, *Sargassum horneri*, sea snots, or marine debris. Therefore, the spectral reflectance dataset presented here is incomplete. The use of data from other similar pathfinders, for example the DLR Earth Sensing Imaging Spectrometer (DESIS) on the ISS (235 bands from 400 – 1000 nm, 30-m resolution, 2018 – present) and the PRecursore IperSpettrale della Missione Applicativa (PRISMA, 237 bands from 400 – 2505 nm, 30-m resolution, 2019 – present), may complement the spectral data using

the same approach (e.g., sea snot reflectance spectra, Hu et al., 2022). Even at its present form, given the large variety
of floating matters presented here, the spectral data may lead to several implications for current and future satellite
missions.
First, although all current multi-band sensors can detect floating matters through their elevated NIR reflectance (Qi et
al., 2020), the Sentinel-3 Ocean and Land Colour Imager (OLCI) appears to be the best to differentiate spectral shapes
in the visible wavelengths because of its 21 spectral bands between 400 and 1,020 nm, especially because of its 620-
nm that can be used to differentiate whether an algae scum appears greenish or brownish, thus providing extra
information to discriminate algae type in the absence of hyperspectral data.
Second, for the same reason, although only 4 bands (blue, green, red, NIR) are available on the PlanetScope (DOVE)
constellation, the recent SuperDOVE constellation is equipped with 4 additional bands with one centered at 610 nm,
thus may significantly enhance the capacity of the current high-resolution sensors (~ 3-4 m or 30 m) in differentiating
greenish and brownish algae types.
Finally, the Ocean Color Instrument (OCI) on NASA's PACE mission, to be launched in 2023, will be the first of its
kind to map global oceans with hyperspectral capacity (5 nm resolution between 340 – 890 nm, plus 7 discrete bands
from 940 to 2260 nm) with a nominal resolution of 1 km. Unlike HICO, OCI will cover global oceans and lakes every
1-2 days, thus providing unprecedented opportunities to detect, differentiate, and quantify various types of floating
matters. The spectral reflectance data, derived from one sensor (HICO) with a stable calibration, may serve as a
consistent dataset to help select the optimal bands towards future applications once PACE data becomes available, for
example, through the use of SAM matrix as demonstrated in Table 1. Likewise, the SBG mission currently being
planned by NASA is expected to have hyperspectral capacity between 380 and 2500 nm with a nominal resolution of
30 m (Cawse-Nicholson et al., 2021); such a mission will provide unprecedented opportunity to map various floating
matters on a global scale where the hyperspectral dataset developed here can help develop algorithms before its launch.
**5. Conclusion**
Through customized atmospheric correction and spectral unmixing, hyperspectral reflectance in the visible and NIR
wavelengths of various floating matters have been derived from HICO measurements over global oceans and lakes.
The reflectance dataset shows distinguishable spectral shapes between floating algae (macroalgae and microalgae)
and non-algae floating matters (*Sargassum fluitans/natans*, *Ulva prolifera*, *kelp*, *Microcystis*, *Trichodesmium*, red
*Noctiluca scintillas*, brine shrimp cysts), and also distinguishable spectral shapes in the visible wavelengths between
different floating algae types. While the approach may be extended to other pathfinder missions to complement the
findings here, the spectral reflectance dataset is expected to help select optimal bands for future hyperspectral satellite
missions to differentiate and quantify the various floating matters in global oceans and lakes.

**Data Availability**

All HICO data used in this analysis are available at the NASA Ocean Biology Distributed Active Archive Center (OB.DAAC, https://oceancolor.gsfc.nasa.gov). The data processing software (SeaDAS) can be obtained from the same source, at https://seadas.gsfc.nasa.gov. The derived HICO spectra in digital data form, as shown in the above figures, are available on-line from the Ecological Spectral Information System (EcoSIS) (http://ecosis.org, doi: 10.21232/74LvC3Kr) (Hu, 2021b).

**Acknowledgements**

This work was supported by the U.S. NASA (NNX17AF57G, 80NSSC21K0422). I thank NASA and the U.S. Naval Research Lab for providing HICO data, thank Lachlan McKinna for providing field-measured reflectance of *Trichodesmium*, and thank Heidi Dierssen for providing field-measured reflectance of whitecaps. Dr. Patrick Launeau and Dr. Qianquo Xing provided useful comments to improve the presentation of this work, whose efforts are appreciated.

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
