# Peer review of "Hyperspectral reflectance spectra of floating matters derived from"

_Earth System Science Data, 2021_

## Author Response (AR1)

======================================================================

Reviewer #1

First, I would like to thank Dr. Launeau for the constructive comments to help justify this work and to better interpret the spectral shapes. Below I address these comments, while the corresponding changes can be found in the revised manuscript.

FRGB is in fact R NIR B color composite image. You could use NIR R B color composition like it is usually done with standard satellite images such as SPOT for instance.

Reply: On the choices of 3 channels to compose RGB: land remote sensing community often used (NIR R B), but then land appears reddish and therefore unnatural. Instead, (R NIR B) will make land appear greenish and also floating vegetation greenish. The NOAA OCView tool (https://www.star.nesdis.noaa.gov/socd/mecb/color/ocview/ocview.html, press "f" on the first page) also used FRGB to inspection of global images. Afterall, these images are used for quick looks to find spatial anomalies, so as long as they can serve for this purpose then either combination should be fine.

I added a reference on the NOAA OCView to further justify the use of FRGB.

The red edge has two origins: the pigment absorption and the cell arrangement.

The pigment absorption is a pure absorption without any reflectance as shown by doi:10.1016/j.jqsrt.2010.08.029 for diatoms. This is only one example. In this case, without reflectance component such microorganisms are transparent in the infrared and must be layering on a reflector to be detected. This has been stressed out in the same work by moving the diatom apart from each other in agarose.  But you are apparently awarded of this effect since you discuss the NIR level of scum in discussion line 243. This is also the case of many other microorganisms like chlorophyte, cyanobacteria and rhodophyte as shown in this other example doi:10.3390/rs10050716

Therefore all pure absorbing pigment distributions of microorganisms cannot be detected in infrared without any reflector at the background like turbid water or scum with reflectance component in near infrared or any other materiel including leaves. This is not the case in the visible spectral range. In your Figure 5 a Monterey, c Taganrog and d both plots are typical shapes of microorganism in NIR absorbing water displaying a pic of reflectance around 700 nm at the end of the pigment absorption and at the beginning of the water absorption as shown in this other example among many others doi.org/10.1016/j.oceano.2017.08.001.

Reply: On the origin of the red-edge reflectance and other reflectance features – I added more descriptions and references, as explained above. This addition should make the paper more informative although its focus is to demonstrate how to derive hyperspectral reflectance from mixed pixels and to provide such derived spectra for community use.

Somehow you agree with this effect since you calculate a SAM in a 450-670 spectral range in Table 1, however missing the deepest absorption feature of the Chlorophyl a at 673 nm. The reflectance plots of the Figure 2 are mainly brown algae characterized by a Chlorophyll c absorption feature at 633 nm with carotenoid while those of the Figure 3 are green algae with Chlorophyll b without carotenoid giving a nice pic of green reflectance at 550 nm. We must wait for line 248 in discussion to discover that a phycocyanin pigment can explain the spectral shape of cyanobacteria… this could have been presented earlier as one of the basic knowledge required for a comprehensive analysis of the results.

On the omission of the 673-nm band in calculating the SAM values: this was done on purpose. The reason is to make it easier to classify different floating algae types. When they form scums or mats, they all show similar red-edge reflectance and therefore similar reflectance trough at 673 nm. Therefore, the inclusion of 673 nm would lead to a lower SAM value, making different types appear more similar (as compared with the same SAM calculation without 673 nm). The justification is already included in the manuscript but I clarified it in both the text and the table caption in the revised manuscript.

On explaining the various pigment-induced reflectance features: I tried to organize results according to the floating matter types (macro, micro, non-living, etc) and explain them sequentially. But I see your point, so I added one paragraph at the end of the Results section to briefly mention the pigment-induced spectral features, while more details are provided in the section below.

So sargassum could be any brown algae and ulva could be any green algae or grass floating on the water… Line 226: "all these floating matters can be differentiated through spectroscopy analysis without any other ancillary information" is probably overstat.

On Sargassum/Ulva versus other brown/green algae: I totally agree that the original statement is an oversell – that type of differentiation only refers to the "endmembers" presented in this paper as opposed to *all* possible endmembers in nature. In the revision this point is well taken, and the sentence is rewritten to clarify.

In fact the discussion of chapter 4 contains the basic knowledge that could have been presented in chapter 2 which could avoid some confusion. All would have been easier to read with a preliminary presentation of the spectral features need for the study from which certain materials and satellite are required.

So I am basically suggesting a reorganization bringing in the front the raison why hyperspectral data are required.

On reorganization chapter 4 to better justify the use of hyperspectral data – this is a good point. Now in the Introduction of the revision, more justifications are presented by citing relevant literature. On the other hand, it's hard to present detailed descriptions of pigment absorption before reporting the spectra, so those descriptions are still kept in chapter 4.

Reviewer #2:

I would like to thank Dr. Qianguo Xing for his constructive comments below. Replies are in blue font.

This is an interesting dataset with a presentation of hyperspectral reflectance of several major floating matters on water surface.  The data and methods presented in this paper are useful for monitoring the aquatic algae, salt shrimp and debris on the basis of space-borne hyperspectral observation.  However, several issues are not clear, and corrections or clarifications may be necessary. Please see my comments below.

Line 38-43:These sentences can be improved. The HICO was desinged for monitoring coastal ocean,  and hyperspectral reflectance of water and non-water targets have already been derived in various applications.

Reply: Yes it is true. The L2 data products contain surface reflectance of water, but these data products are not applicable for floating matters that only occupy a small portion of an image pixel. This is why the customized processing is used in this paper. I rewrote this paragraph and last paragraph to clarify when customized atmospheric correction and pixel unmixing are required even though NASA already has surface reflectance data products.

Line 40 and Line 53:"9,411 scenes" may be the most part of the images collected during the mission of HICO, but not "all".  Please check the following reference and my next comments.  Reference: https://oceancolor.gsfc.nasa.gov/hico/

Reply: Yes it is true that HICO has collected > 10,000 scenes, but of these, only 9,411 are available through https://oceancolor.gsfc.nasa.gov. I clarified these two numbers in this paragraph and in the data section.

The blooms of Mesodinium rubrum were mapped by HICO.  It would be useful to check the possibility of the differentiation between the Mesodinium rubrum and the red NS on the basis of reflectance.   Reference: Dierssen et al., 2015. Space station image captures a red tide ciliate bloom at high spectral and spatial resolution.    https://www.ncbi.nlm.nih.gov/pmc/articles/PMC4672822/

Reply: This is a good suggestion. I actually searched for Mesodinium cases including this one, but couldn't find any algae scum from any cases. I doubled checked the Dierssen et al. case but still couldn't find any algae scum (I attach two images below and in a pdf file). So these are not floating algae and therefore not included in the paper. In Dierseen et al (2015), they emphasized the CDOM and pigment fluorescence in the visible wavelengths as opposed to NIR wavelengths due to algae scums. But I agree this is a good discussion point so I added several sentences to discuss.

As shown by the title of this paper, the reflectance of floating matters was derived and compared. However, for the kelp,  as mentioned in line 235,  it is usually not floating in the sea surface. So, why not consider the effects of the emerged portion of macroalgae? As the discussed in lines 183-190, this may be one of the major reasons causing the spectral difference (in reflectance or SAM) between Sargassum and kelp. The sargassum and kelp can be emerged or submerged, so I suggest to make a clarification that the sargassum in this paper refers to the specific floating sargassum species.  For sargassum, different terms are used and may cause confusions: "Sargassum", " Sargassum fluitans/natans", "pelagic Sargassum," and  "Sargassum honeric".  Actually, in most cases,  Sargassum honeric is fixed to sea bed and grows under surbmerged conditions.

Reply: These are great points. Depending on the submerged depth, the red-edge signal of kelp may vary a lot, leading to large uncertainties during pixel unmixing. I clarified this in the vision. Some terms (e.g., Sargassum, Noctiluca) are used in the Introduction to refer to the general type, but the derived spectra are for more specific types. I clarified this in Section 4.2 in the revision. For the entire HICO archive, I couldn't find a single case for S. horneri (line 297 in the original manuscript).

[Figure]

RGB and FRGB HICO images on 9/23/2012 over the western Long Island Sound (WLIS) showing no surface scums although a Mesodinium bloom has been reported (Dierssen et al., 2015).